# Retinal Vascular Changes in Heart Failure with Preserved Ejection Fraction Using *Optical Coherence Tomography Angiography*

**DOI:** 10.3390/jcm13071892

**Published:** 2024-03-25

**Authors:** Jerremy Weerts, Anne G. Raafs, Birgit Sandhoefner, Frank C. T. van der Heide, Sanne G. J. Mourmans, Nicolas Wolff, Robert P. Finger, Peyman Falahat, Maximilian W. M. Wintergerst, Vanessa P. M. van Empel, Stephane R. B. Heymans

**Affiliations:** 1Department of Cardiology, Cardiovascular Research Institute Maastricht (CARIM), Maastricht University Medical Centre+ (MUMC+), P.O. Box 616, 6200 MD Maastricht, The Netherlands; a.raafs@maastrichtuniversity.nl (A.G.R.); sanne.mourmans@mumc.nl (S.G.J.M.); vanessa.van.empel@mumc.nl (V.P.M.v.E.); 2Carl ZEISS Meditec Inc., 5300 Central Parkway, Dublin, CA 94568, USAnicolas.wolff@zeiss.com (N.W.); 3Department of Internal Medicine, Cardiovascular Research Institute Maastricht (CARIM), Maastricht University Medical Centre+ (MUMC+), 6200 MD Maastricht, The Netherlands; frank.vander.heide@mumc.nl; 4University Eye Clinic Maastricht, Maastricht University Medical Centre+ (MUMC+), 6200 MD Maastricht, The Netherlands; 5MHeNS, School for Mental Health and NeuroScience, Maastricht University, 6200 MD Maastricht, The Netherlands; 6Department of Ophthalmology, University Hospital Bonn, 53127 Bonn, Germany; robert.finger@ukbonn.de (R.P.F.); peyman.falahat@ukbonn.de (P.F.); maximilian.wintergerst@ukbonn.de (M.W.M.W.); 7Department of Cardiovascular Research, University of Leuven, 3000 Leuven, Belgium

**Keywords:** heart failure with preserved ejection fraction, OCT-A, LV diastolic dysfunction, retina alterations, microvascular density, imaging biomarker

## Abstract

**Background:** Systemic microvascular regression and dysfunction are considered important underlying mechanisms in heart failure with preserved ejection fraction (HFpEF), but retinal changes are unknown. **Methods:** This prospective study aimed to investigate whether retinal microvascular and structural parameters assessed using optical coherence tomography angiography (OCT-A) differ between patients with HFpEF and control individuals (i.e., capillary vessel density, thickness of retina layers). We also aimed to assess the associations of retinal parameters with clinical and echocardiographic parameters in HFpEF. HFpEF patients, but not controls, underwent echocardiography. Macula-centered 6 × 6 mm volume scans were computed of both eyes. **Results:** Twenty-two HFpEF patients and 24 controls without known HFpEF were evaluated, with an age of 74 [68–80] vs. 68 [58–77] years (*p* = 0.027), and 73% vs. 42% females (*p* = 0.034), respectively. HFpEF patients showed vascular degeneration compared to controls, depicted by lower macular vessel density (*p* < 0.001) and macular ganglion cell-inner plexiform layer thickness (*p* = 0.025), and a trend towards lower total retinal volume (*p* = 0.050) on OCT-A. In HFpEF, a lower total retinal volume was associated with markers of diastolic dysfunction (septal e’, septal and average E/e’: *R*^2^ = 0.38, 0.36, 0.25, respectively; all *p* < 0.05), even after adjustment for age, sex, diabetes mellitus, or atrial fibrillation. **Conclusions:** Patients with HFpEF showed clear levels of retinal vascular changes compared to control individuals, and retinal alterations appeared to be associated with markers of more severe diastolic dysfunction in HFpEF. OCT-A may therefore be a promising technique for monitoring systemic microvascular regression and cardiac diastolic dysfunction.

## 1. Introduction

Heart failure with preserved ejection fraction (HFpEF) is a complex syndrome and is associated with high mortality rates and a poor quality of life [1]. Patients with HFpEF are often older, predominantly female, and have an increased incidence of cardiovascular comorbidities such as diabetes mellitus (DM), hypertension, renal dysfunction, obesity, and atrial fibrillation [1]. A systemic pro-inflammatory state causing microvascular endothelial regression (loss of capillary density) and dysfunction is considered an early mark of elevated cardiovascular risk, systemic organ damage, and HFpEF development [1]. Capillary regression may occur in parallel in the heart, eye, kidney, skin, skeletal muscles, and brain in response to hypertension, DM, dyslipidaemia, and ageing [2], the same comorbidities associated with HFpEF development.

Imaging the eye allows a direct view of the retinal arterial and capillary bed and may therefore represent an ideal window to systemic cardiovascular diseases and its microvascular regression and dysfunction. Studies addressing the retinal microvascular components in HFpEF, however, are lacking.

Optical coherence tomography angiography (OCT-A) is a novel non-invasive and time-efficient imaging technique which allows one to visualize and subsequently quantify the retinal capillary vessel density, along with certain layer’s thicknesses such as those rich in neurons [3,4]. OCT-A uses laser-light reflectance of the surface of moving erythrocytes up to a histological level to accurately depict vessels and capillaries through different segmented layers of the retina and choroid, thereby eliminating the need for intravascular dyes [3]. With OCT-A, microvascular regression and dysfunction can be estimated from lower vessel density (loss of macular superficial and deep capillaries), a greater foveal avascular zone (FAZ; greater loss of perifoveal capillaries), and reduced blood flow in retinal arterioles (perfusion density) [3].

The aim of this study was to identify differences in the retinal microvascular vessel density, FAZ, perfusion density, and ganglion cell-inner plexiform layer (GCIPL) thickness in patients with HFpEF versus control individuals. Subsequently, the above retinal OCT-A parameters were related to clinical and echocardiographic parameters of diastolic dysfunction in patients with HFpEF.

## 2. Methods

### 2.1. Design and Participants

The present study is part of a prospective observational study on the peripheral microcirculation in HFpEF (Netherlands Trial Register NL7655), approved by the ethics committees azM/UM, the Netherlands (ethics approval ID 19-005), and is in accordance with the Helsinki declaration. Clinical data and echocardiography were obtained from standardized clinical care at a dedicated HFpEF outpatient clinic at the Maastricht University Medical Centre+, The Netherlands [5]. HFpEF was defined according to the European Society of Cardiology heart failure 2016 guidelines [6], requiring patients to have a left ventricular ejection fraction of 50% or higher. Diagnosis was made by consensus of at least two heart failure specialists. Sample size was limited to patients presenting for the NL7655 study from 31 October 2019 to 6 December 2019. Ocular pathologies were identified based on medical history, intra-ocular pressures, and retinography acquisitions (assessed while blinded for other clinical data). Anonymous data from control individuals who underwent OCT-A measurements were obtained from a former case–control study at the ophthalmology outpatient clinic at the University Hospital Bonn, Germany (ethics approval ID 047/18) [7]. Healthy control individuals without peripheral artery disease were selected based on medical records and a self-reported assessment. They had no history or recent clinical suspicion for heart failure and had no clinical indication for echocardiography. Moreover, control individuals had no current ocular symptoms, history of ocular surgery or ocular diseases (except cataract surgery), or presence of DM. All participants gave written informed consent.

### 2.2. OCT-A Measurements

OCT-A was performed in all participants using the ZEISS PLEX Elite 9000 (Carl Zeiss Meditec, Inc., Dublin, CA, USA), imaging macula-centered volume scans of 6 × 6 mm in both eyes. For quantification analysis, anonymized imaging files were exported and uploaded in the Advanced Retina Imaging network (ARI, Carl Zeiss Meditec, Inc., Dublin, CA, USA) [8]. Macular vessel density and parameters of FAZ geometry (area, perimeter, and tortuosity) were assessed from the complete 6 × 6 mm image at the level of the superficial and deep capillary plexus. The superficial capillary plexus layer was located between the internal limiting membrane and the inner plexiform layer, and the deep layer between the inner plexiform layer and the outer plexiform layer) (ARI macular density v0.7.2). Layer thickness parameters were derived from B-scans of the PLEX Elite 9000 6 × 6 mm acquisitions (ARI retina thickness v0.1). GCIPL thickness was assessed from GCL to outer border of inner plexiform layer, and average retinal thickness and total volume from the internal limiting membrane to the retinal pigment (ARI GCIPL analysis v0.2). For each participant, the median of measurements from both eyes was used for statistical analyses. In case of inadequate image or processing quality in an acquisition, only the results from acquisitions with sufficient quality were used for that patient. Acquisition quality was scored similar to the OSCAR-IB criteria [9], while also taking into account more OCT-A specific artefacts (i.e., motion or projection). In addition, post-computation quality check was performed, including alignment of the quantification grid verification, vessel dropout after binarization of the image assessment and segmentation challenges assessment.

### 2.3. Echocardiographic Analysis

All patients with HFpEF underwent comprehensive 2-dimensional echocardiographic imaging, including Doppler and tissue Doppler imaging, using commercially available ultrasound systems with harmonic imaging as previously described [10]. Briefly, all measurements were performed by experienced sonographers as part of routine clinical care, blinded for OCT-A results, and in accordance with the American Society of Echocardiography and the European Association of Cardiovascular Imaging guidelines [11]. During the echocardiography acquisitions, dedicated non-foreshortened apical recordings were obtained to assess left ventricular and left atrial morphology and function. Tissue Doppler e’ velocities were measured at the mitral annulus’s septal and lateral aspects with optimized sample volume and placement. A cardiologist with echocardiography expertise verified all analyses.

### 2.4. Statistical Analysis

Variables are displayed as numbers (percentage) and median [interquartile range (IQR)], as appropriate. Differences between patients with HFpEF and control individuals were statistically tested using the Chi-square or Mann–Whitney U test, depending on the data type. Normality of variables was assessed visually using normal P–P plots and histograms. Initially, we performed unadjusted linear regression analyses to investigate the relationship between OCT-A and echocardiographic parameters in patients with HFpEF. The significant associations were further evaluated using multivariable linear regression, adjusted for age, sex, DM, and atrial fibrillation separately. DM was chosen because of its known impact on microvascular function, and atrial fibrillation because it is a well-known important prognostic factor in HFpEF. We did not include all variables into one model for reasons of statistical power, to ensure a maximum amount of 10% of the sample size as covariate. Interaction analyses between the independent and dependent variable of each model was performed to assess effect modification. Subsequently, we assessed correlations between OCT-A markers that were different between patients with HFpEF and control individuals using linear regression. Finally, all linear regression analyses were performed for the left and right eye separately, and in patients without eye diseases. Missing data were handled with pairwise deletion. Due to ethical and regulatory restrictions, clinical data from control individuals were analyzed anonymously and separately from OCT-A acquisitions, precluding linear regression analyses between clinical data and retinal markers in this group. A two-sided *p*-value of <0.05 was considered significant. Statistical analysis was performed using SPSS 26.0 (IBM Corp., Armon, NY, USA) software.

## 3. Results

Twenty-two consecutively recruited HFpEF and 24 control patients were included. The clinical characteristics of both groups are described in Table 1. Patients with HFpEF were more often female and slightly older. They had cardiovascular comorbidities such as atrial fibrillation, DM, hypertension, and dyslipidaemia more often.

### 3.1. Retinal Differences in HFpEF

Macular vessel density and GCIPL, reflecting the capillary density and neuronal layer thickness, were significantly decreased in patients with HFpEF as compared to control individuals (Figure 1, Table 1). We also found a directionally similar trend, although it was not statistically significant (*p* = 0.050), towards a lower total retinal volume in patients with HFpEF.

In addition, in patients with HFpEF a lower total retinal volume was associated with worse septal e’ and septal and average E/e’ (Figure 2, Table 2). These associations remained significant after adjustment for age, sex, DM, or atrial fibrillation. No effect modification was observed between these variables and retinal parameters. All associations with total retinal volume were somewhat stronger after adjustment for sex.

### 3.2. Associations between OCT-A Markers

Patients with HFpEF and control individuals showed different associations between OCT-A markers. The macular vessel density was associated with the FAZ perimeter and circulatory index in patients with HFpEF (*R*^2^ = 0.272 and 0.301, *p* = 0.018 and 0.012, respectively), but not in control individuals (*R*^2^ = 0.142 and 0.003, *p* = 0.083 and 0.806, respectively). In addition, macular GCIPL thickness was associated with central retinal thickness and total volume in patients with HFpEF (*R*^2^ = 0.197 and 0.257, *p* = 0.038 and 0.016, respectively), and even more so in control individuals (*R*^2^ = 0.230 and 0.773, *p* = 0.021 and <0.001, respectively).

### 3.3. Asymmetry and Impact of Ocular Pathologies

Remarkably, correlations were mostly due to alterations in the right eyes of patients (Table 3 and Table 4). In line herewith, a larger FAZ area and perimeter in the right eye of patients with HFpEF had a correlation trend with increased left atrial volume index (LAVI) (n = 13, *R*^2^ = 0.248 and 0.283 with *p* = 0.083 and 0.061, respectively) but not in the left eye (n = 16, *R*^2^ = 0.080 and 0.030 with *p* = 0.288 and 0.524, respectively).

Four patients with HFpEF had relevant ocular pathologies: two (9.1%) had glaucoma in the right eye, one (4.5%) had glaucoma in both eyes, and one (4.5%) had diabetic retinopathy in the left eye. The results of all comparative and regression analyses remained similar, including their significance and asymmetrical differences, when patients with ocular pathologies were excluded.

## 4. Discussion

To the best of our knowledge, this study is the first to investigate the associations of OCT-A-assessed retinal vascular changes and HFpEF. First, we found that patients with HFpEF had a lower macular vessel density and GCIPL thickness, and a trend towards a lower total retinal volume, as compared to the control group. Second, in patients with HFpEF, a lower total retinal volume was significantly associated with septal E/e’ and e’, but not left ventricular (LV) mass index, TR speed, or NT-proBNP in patients with HFpEF. These findings suggest alterations in both the retinal capillaries and neurons in HFpEF, and possibly more so in those with more severe diastolic dysfunction or higher left-sided cardiac filling pressures. This novel topic has been understudied in HFpEF populations, yet these preliminary results could matter a great deal for future research.

This study’s findings are in line with findings from previous studies, and extend the current knowledge by including detailed retinal measurements. Impaired retinal arteriolar and venular reactiveness has been associated with cardiovascular risk factors and HF with reduced ejection fraction using dynamic vessel analysis (DVA) [12]. Although DVA does not visualize smaller capillaries or microvascular components, such as density or flow, it suggests retinal vascular alterations in HF that may also occur in HFpEF.

Our findings suggest distinct retinal changes in both the vascular and neuronal foveal components in patients with HFpEF. The lower macular vessel density found in patients with HFpEF in the current study is in line with prior HFpEF studies in other vascular beds of organs and extremities, showing more microvascular regression and less perfusion as signs of systemic microvascular dysfunction in HFpEF [13]. Rather than only impaired perfusion suggested by a lower perfusion density or only vascular regression suggested by a larger FAZ, it is likely that the combination of both these retinal impairments in patients with HFpEF reflect a lower macular vessel density.

Moreover, we reported an impaired GCIPL thickness in HFpEF. GCIPL thickness did not correlate with retinal vessel density, which was in line with previous findings [14]. However, GC thickness is known to be correlated with its more superficial retinal nerve fiber layer (RNFL) [15]. Although RNFL thickness was not associated with classic cardiovascular risk factors [16], it has been associated with HF in patients with an LV ejection fraction <55% [17], indicating thinner nerve layers in patients with more severe HF. Thinner retinal neuronal layers may also indicate the development of more systemic neuropathy or systemic organ damage, as reported in DM patients [18].

The present findings are consistent with the hypothesis of HFpEF as a systemic syndrome accompanied by microvascular dysfunction. Microvascular endothelial dysfunction in the heart of patients with HFpEF can cause a cascade of events, resulting in concentric LV remodeling, diastolic dysfunction, and higher end-systolic volume and pressure [1], reflected by non-invasive echocardiography markers such as decreased e’ and elevated E/e’ [19]. The degree of impairment in these echocardiography markers is associated with a worse prognosis [20,21]. These impairments can be improved, thereby partially restoring exercise capacity [22]. Therefore, the association between these markers and a smaller total retinal volume in HFpEF, even after correction for demographics and comorbidities, suggests retinal alterations may relate to disease severity in HfpEF and may imply systemic consequences of the syndrome. The total retinal volume likely reflects a sum of subtle changes of vascular regression in HFpEF, but which specific retinal alterations are exaggerated in patients with HFpEF and become worse during the progression of the disease, extending to classical metabolic risk factors, warrants further studies. More broadly, the prognostic relevance of retinal disease has been shown in patients with HFpEF and DM, in which self-reported retinopathy was associated with future heart failure hospitalizations and higher mortality [23].

The asymmetric retinal alterations and associations found in the present study, which were more prominent in the right eye of patients with HFpEF, were unexpected findings and demand cautious interpretation given our sample size. Possible hypotheses for this retinal asymmetry require more specific data, and may include effects of asymmetry in left to right blood pressures [24], asymmetrical carotid atherosclerosis and blood flow [25], or eye dominance [26].

Our findings are consistent with the concept that it may be possible to monitor microvascular regression and diastolic dysfunction of the heart via the retina, suggesting that functional and structural retinal changes occur in patients with HFpEF during the progression of the disease. More longitudinal research on retinal microvascular alterations with OCT-A is warranted to explore its value as a non-invasive biomarker for early HFpEF development, HFpEF progression, and response to therapy.

The present study has certain strengths and limitations. We employed state-of-the art and highly accurate measurement techniques to assess retinal vascular differences between well-phenotyped patients with HFpEF and control individuals. We also adjusted the associations analyzed within the HFpEF group for multiple confounders, reducing the risk of finding spurious associations due to the effects of these confounders. The small sample size and the lack of echocardiographic data of the control population limit the study findings. Although control individuals were considered healthy, the absence of a clinical presentation assessment and echocardiography cannot exclude the presence of stage B (pre-)heart failure in control individuals and restrained comparative group analyses adjusted for confounders. However, the presence of any heart failure stage in control individuals would likely have attenuated rather than increased the observed group differences. In future studies, echocardiography should be employed in each study group to ensure the specificity of echocardiographic–OCT-A associations per patient group. Age differences between patients with HFpEF and control individuals may have been partially due to the group differences (median age difference of six years), but the retinal differences between groups exceeded expected changes based on reference data for subjects with ages ranging from 30 to 80 years [27]. This is supported by an observational study that reported, for example, a loss of 0.008 mm^3^ total retinal volume per year [28], which is a hundredfold smaller than our observed group difference of 0.8 mm^3^. The higher prevalence of females within the HFpEF group compared to control individuals may also have been due to differences in retinal marker results, although sex-dependent differences are reported to be less prominent [7,28,29] than the differences between HFpEF and control individuals we observed in the present study. Moreover, sex did not impact the associations we observed between retinal and cardiac diastolic dysfunction markers in patients with HFpEF, suggesting that the retinal alterations are relevant in both sexes. Finally, the cross-sectional design of the study precludes conclusions on the temporality of the associations. Longitudinal data on microvascular alterations, the degree of diastolic dysfunction, and elevated filling pressures are required to evaluate how these processes influence each other.

In conclusion, patients with HFpEF showed clear levels of retinal vascular changes compared to control individuals, and retinal alterations appeared to be associated with HFpEF severity. OCT-A of the eye may thus be a promising non-invasive technique for the monitoring of systemic microvascular regression and diastolic dysfunction of the heart.

## Figures and Tables

**Figure 1 jcm-13-01892-f001:**
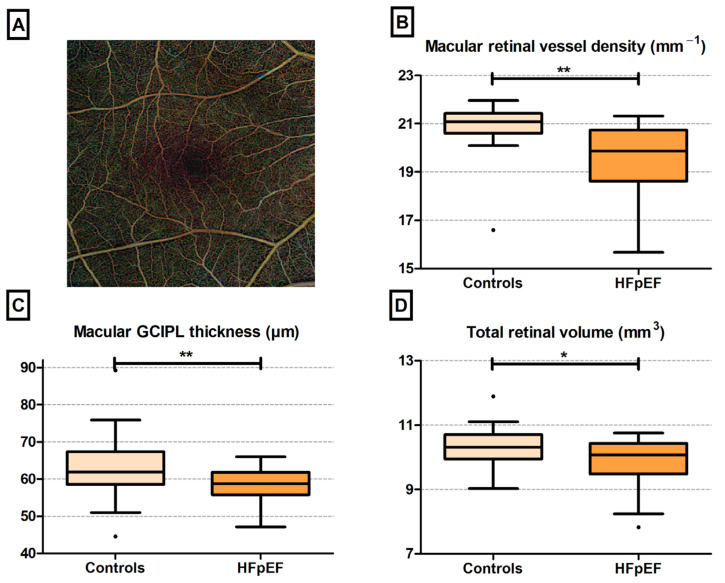
OCT-A-derived retinal alterations in patients with HFpEF compared to control individuals. (**A**) Example of OCT-A image of a female patient with HFpEF, showing a 6 × 6 mm montage image; (**B**–**D**) OCT-A-derived retinal alterations in patients with HFpEF compared to control individuals. Dots within the bar plots indicate outliers. ** *p* < 0.05; * *p* = 0.05.

**Figure 2 jcm-13-01892-f002:**
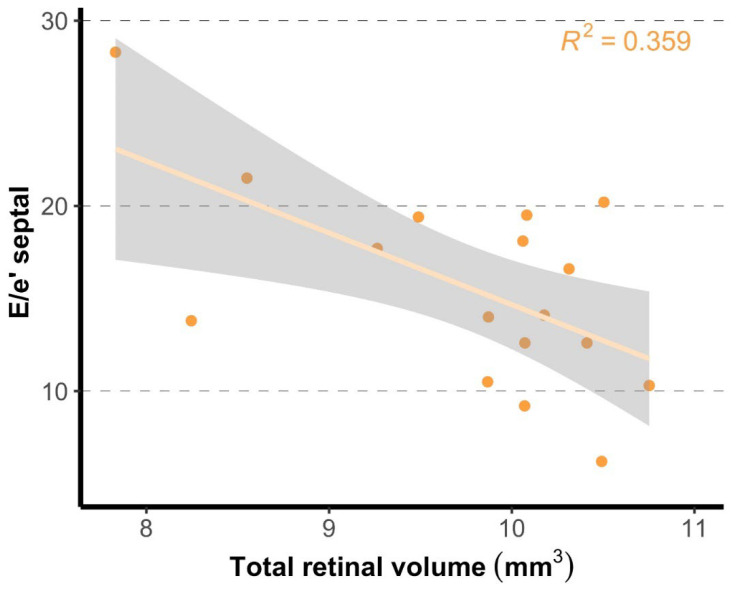
Association of total retinal volume and septal E/e’ in patients with HFpEF. A higher E/e’ was associated with a lower total retinal volume.

**Table 1 jcm-13-01892-t001:** Clinical characteristics of the study population.

	HFpEF n = 22	Controls n = 24	*p*-Value
Age, years	**74 [68–80]**	**68 [58–77]**	**0.027**
Female sex, n (%)	**16 (73)**	**10 (42)**	**0.034**
**Cardiovascular comorbidities, n (%)**			
Atrial fibrillation	**12 (55)**	**2 (8)**	**<0.001**
Diabetes mellitus type 2	**6 (27)**	**0 (0)**	**0.008**
Hypertension	**18 (82)**	**12 (50)**	**0.024**
Dyslipidaemia	**14 (64)**	**3 (13)**	**<0.001**
Stroke	**5 (23)**	**0 (0)**	**0.019**
**Clinical presentation**			
Diastolic blood pressure, mmHg	81 [66–89]	-	
Systolic blood pressure, mmHg	141 [124–151]	-	
NT-proBNP, pg/mL	553 [302–1277]	-	
BMI, kg/m^2^	29 [25–37]	-	
**Echocardiography**			
LVEF, %	62 [58–66]	-	
LVMI, g/m^2^	72 [60–86]	-	
e’ septal, cm/s	6.0 [5.3–7.8]	-	
e’ lateral, cm/s	7.9 [6.2–10.2]	-	
E/e’ septal	14.1 [11.6–19.5]	-	
E/e’ lateral	12.5 [9.3–15.9]	-	
E/e’ average	13.6 [10.7–17.8]	-	
TR speed, cm/s	2.6 [2.3–3.0]	-	
RVSP, mmHg	32.5 [25.0–41.3]	-	
LAVI, ml/m^2^	48 [39–55]	-	
**OCT angiographic measurements**			
Central retinal perfusion density	0.22 [0.19–0.32]	0.26 [0.21–0.31]	0.525
Central retinal vessel density, mm^−1^	10.8 [9.2–15.7]	12.9 [10.5–15.4]	0.540
Macular vessel density, mm^−1^	**19.9 [18.6–20.7]**	**21.1 [20.6–21.4]**	**<0.001**
Macular GCIPL thickness, µm	**58.8 [55.8–61.8]**	**62.0 [58.6–67.4]**	**0.025**
Average central retinal thickness, µm	281 [257–296]	279 [263–286]	0.668
Total retinal volume, mm^3^	**9.5 [10.1–10.4]**	**10.3 [9.9–10.7]**	**0.050**
FAZ perimeter, mm	1.92 [1.48–2.18]	1.98 [1.67–2.39]	0.632
FAZ area, mm^2^	0.23 [0.13–0.29]	0.23 [0.16–0.28]	0.743
FAZ circularity index	0.76 [0.71–0.80]	0.71 [0.70–0.78]	0.279

Legend: Bold numbers indicates statistically significant group differences. BMI = body mass index, FAZ = foveal avascular zone, GCIPL = ganglion cell and inner plexiform layer, HFpEF = heart failure with preserved ejection fraction, LAVI = left atrial volume index, LVEF = left ventricular ejection fraction, LVMI = left ventricular mass index, NT-proBNP = N-terminal-pro hormone B-type natriuretic peptide, RVSP = right ventricular systolic pressure, TR = tricuspid regurgitation.

**Table 2 jcm-13-01892-t002:** Associations of lower total retinal volume with cardiac parameters in HFpEF.

	Β	95%CI	SE β	Standardized β	*p*-Value	*R* ^2^
**E/e’ septal**	**0.09**	**0.03–0.16**	**0.03**	**0.60**	**0.011**	**0.36**
Adjusted for age	0.09	0.02–0.16	0.03	0.59	0.015	0.37
Adjusted for sex	0.11	0.05–0.17	0.03	0.70	0.002	0.53
Adjusted for DM	0.10	0.04–0.17	0.03	0.66	0.013	0.47
Adjusted for AF	0.10	0.02–0.18	0.04	0.67	0.017	0.37
**e’ septal (log(cm/s))**	**−3.42**	**−5.74–−1.10**	**1.09**	**−0.62**	**0.006**	**0.38**
Adjusted for age	−3.41	−5.81–−1.00	1.13	−0.61	0.009	0.38
Adjusted for sex	−3.73	−6.07–−1.40	1.10	−0.67	0.004	0.44
Adjusted for DM	−3.32	−5.68–−0.96	1.11	−0.60	0.009	0.41
Adjusted for AF	−3.78	−6.62–−0.95	1.33	−0.68	0.012	0.39
**E/e’ average**	**0.09**	**0.004–0.17**	**0.04**	**0.50**	**0.041**	**0.25**
Adjusted for age	0.09	0.006–0.18	0.04	0.52	0.038	0.29
Adjusted for sex	0.12	0.04–0.20	0.04	0.66	0.008	0.45
Adjusted for DM	0.10	0.01–0.18	0.04	0.54	0.030	0.33
Adjusted for AF	0.09	−0.01–0.20	0.05	0.52	0.067	0.25

Legend: A positive beta indicates a lower total retinal volume on a continuous scale. No effect modification was observed between the adjusting factors and the independent variables. Bold text indicates unadjusted analyses. AF = atrial fibrillation, HFpEF = heart failure with preserved ejection fraction, DM = diabetes mellitus.

**Table 3 jcm-13-01892-t003:** OCT angiography findings per eye.

OCT Angiographic Measurements	HFpEF n = 22	Controls n = 24	*p*-Value
Central retinal perfusion density	0.22 [0.19–0.32]	0.26 [0.21–0.31]	0.525
OS (n = 21/15)	0.25 [0.18–0.33]	0.26 [0.18–0.31]	0.680
OD (n = 18/18)	0.24 [0.20–0.31]	0.25 [0.21–0.31]	0.606
Central retinal vessel density, mm^−1^	10.8 [9.2–15.7]	12.9 [10.5–15.4]	0.540
OS (n = 21/15)	11.8 [9.3–16.0]	13.0 [8.9–15.4]	0.849
OD (n = 18/18)	11.8 [8.5–15.3]	12.4 [10.4–14.9]	0.650
Macular vessel density, mm^−1^	**19.9 [18.6–20.7]**	**21.1 [20.6–21.4]**	**<0.001**
OS (n = 21/15)	**20.3 [18.8–20.8]**	**20.8 [20.5–21.5]**	**0.009**
OD (n = 18/18)	**20.0 [17.8–20.9]**	**21.0 [20.8–21.3]**	**0.002**
Macular GCIPL thickness, µm	**58.8 [55.8–61.8]**	**62.0 [58.6–67.4]**	**0.025**
OS (n = 20/15)	60.7 [56.0–63.8]	61.1 [58.6–64.9]	0.191
OD (n = 19/18)	**58.2 [53.7–60.1]**	**62.2 [59.8–67.4]**	**0.004**
Average central retinal thickness, µm	281 [257–296]	279 [263–286]	0.668
OS (n = 19/15)	284 [265–299]	270 [250–280]	0.111
OD (n = 20/19)	287 [256–302]	282 [261–286]	0.647
Total retinal volume, mm^3^	9.5 [10.1–10.4]	10.3 [9.9–10.7]	0.050
OS (n = 19/15)	10.2 [9.8–10.4]	10.3 [10.0–10.6]	0.354
OD (n = 20/19)	**10.0 [9.5–10.4]**	**10.5 [9.9–10.9]**	**0.038**
FAZ perimeter, mm	1.92 [1.48–2.18]	1.98 [1.67–2.39]	0.632
OS (n = 17/15)	1.94 [1.52–2.38]	2.05 [1.85–2.48]	0.295
OD (n = 14/17)	1.74 [1.23–2.10]	1.99 [1.66–2.35]	0.215
FAZ area, mm^2^	0.23 [0.13–0.29]	0.23 [0.16–0.28]	0.743
OS (n = 17/15)	0.23 [0.14–0.30]	0.24 [0.21–0.38]	0.551
OD (n = 14/17)	0.17 [0.09–0.27]	0.23 [0.16–0.30]	0.316
FAZ circularity index	0.76 [0.71–0.80]	0.71 [0.70–0.78]	0.279
OS (n = 17/15)	0.77 [0.71–0.81]	0.70 [0.66–0.78]	0.216
OD (n = 14/17)	0.75 [0.73–0.78]	0.72 [0.70–0.77]	0.262

Legend: Number per eye indicates subjects with HFpEF and control individuals, respectively. Bold numbers indicate statistically significant group differences. GCIPL = ganglion cell and inner plexiform layer, FAZ = foveal avascular zone, OD = right eye, OS = left eye.

**Table 4 jcm-13-01892-t004:** Associations of lower total retinal volume with cardiac parameters in HFpEF per eye.

	Left Eye				Right Eye			
	β	St. β	*p*-Value	*R* ^2^	β	St. β	*p*-Value	*R* ^2^
**E/e’ septal**	**0.02**	**0.112**	**0.680**	**0.013**	**0.17**	**0.687**	**0.003**	**0.471**
Adjusted for age	0.02	−0.190	0.496	0.049	0.17	0.685	0.005	0.474
Adjusted for sex	0.05	0.249	0.349	0.220	0.17	0.720	0.004	0.491
Adjusted for diabetes mellitus	0.04	0.185	0.507	0.101	0.17	0.706	0.004	0.488
Adjusted for atrial fibrillation	−0.01	−0.055	0.872	0.063	0.19	0.804	0.003	0.518
**e’ septal (log(cm/s))**	**−1.12**	**−0.168**	**0.519**	**0.028**	**−5.77**	**−0.677**	**0.003**	**0.458**
Adjusted for age	−0.03	−0.153	0.543	0.052	−5.77	−0.677	0.004	0.458
Adjusted for sex	−1.66	−0.250	0.340	0.149	−5.82	−0.682	0.004	0.459
Adjusted for diabetes mellitus	−0.96	−0.144	0.580	0.098	−5.80	−0.680	0.004	0.459
Adjusted for atrial fibrillation	−0.12	−0.019	0.953	0.073	−6.89	−0.808	0.003	0.502
**E/e’ average**	**0.02**	**0.084**	**0.757**	**0.007**	**0.16**	**0.568**	**0.022**	**0.322**
Adjusted for age	0.02	0.105	0.706	0.049	0.16	0.583	0.023	0.340
Adjusted for sex	0.06	0.267	0.327	0.225	0.17	0.626	0.020	0.352
Adjusted for diabetes mellitus	0.03	0.132	0.633	0.085	0.16	0.577	0.025	0.329
Adjusted for atrial fibrillation	−0.02	−0.071	0.828	0.065	0.18	0.638	0.026	0.341

Legend: A positive beta indicates a lower total retinal volume on a continuous scale. Bold text indicates unadjusted analyses. St. β = standardized β.

## Data Availability

All relevant data are within the paper. The data underlying this article may be shared upon reasonable request to the corresponding authors.

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
