# Peer review of "Retinal Vascular Changes in Heart Failure with Preserved Ejection Fraction Using Optical Coherence Tomography Angiography"

_jcm, 2024, doi:10.3390/jcm13071892_

Round 1

Reviewer 1 Report

Comments and Suggestions for Authors

Key word:

Please rephrase two key words heart failure and preserved ejection fraction in one: Heart failure with preserved ejection fraction

Methodology

I have a significant methodological consideration with respect to this study. The fact that controls do not have data about clinical presentation and echocardiography is significant shortcoming of this study, despite the explanation given by authors.

However, from the ethical point of view, authors do mention all these shortcomings as study limitations.

The sample size is too small to enable adjustments for age, sex, diabetes, AF etc. Further one, males predominate, something that is opposite to actual gender distribution of HFpEF.

With respect to age there is statistically significant difference in the mean age between HFpEF patients and controls, which can bring the question of registered differences simply as a result of age difference between study subject and controls. Also, other issue is missing data on clinical presentation for control patients, so one cannot exclude the possible influence of hypertension (present in 50% of control patients), or the obesity in controls. Same goes for echocardiography parameters. Having this data would make OCT data more reliable.

However, despite methodology shortcomings this study has clinical relevance.

Clinical significance of this manuscript:

Imaging the eye allows a direct view on the retinal arterial and capillary bed and may 56 therefore represent an ideal window to systemic cardiovascular diseases and its microvascular regression and dysfunction. Studies addressing the retinal microvascular components in HFpEF, however, are lacking.

Author Response

  1. Please rephrase two key words heart failure and preserved ejection fraction in one: Heart failure with preserved ejection fraction

Thank you. The keyword has changed accordingly.

  1. I have a significant methodological consideration with respect to this study. The fact that controls do not have data about clinical presentation and echocardiography is significant shortcoming of this study, despite the explanation given by authors. However, from the ethical point of view, authors do mention all these shortcomings as study limitations.

We acknowledge your valuable comment. Although the study design of this (proof-of-principle) study itself cannot be amended, more clarity for the group differences was warranted and has been added to the manuscript:
Healthy control individuals without peripheral artery disease were selected based on medical records and a self-reported assessment. They had no history or recent clinical suspicion for heart failure and had no clinical indication for echocardiography. Moreover, control individuals had no current ocular symptoms, history of ocular surgery or ocular diseases (except cataract surgery), or presence of DM.“ (line 92-96)

Even without echocardiography controls would not fulfill the clinical definition of heart failure (stage C or D). We also extended the limitations of our manuscript: “Although control individuals were considered healthy, the absence of clinical presentation assessment and echocardiography cannot exclude the presence of stage B (pre-)heart failure in control individuals and restrained comparative group analyses adjusted for confounders. However, the presence of any heart failure stage in control individuals would likely have attenuated rather than increased the observed group differences. In future studies, echocardiography should be employed in each study group to ensure the specificity of echocardiographic – OCT-A associations per patient group. (line 275-282)

  1. The sample size is too small to enable adjustments for age, sex, diabetes, AF etc. Further one, males predominate, something that is opposite to actual gender distribution of HFpEF.

We agree the sample size is small and multiple adjustments should be performed with caution. As such, we performed only single adjustments for associations per clinical variable to meet the generally accepted 10% sample size rule (i.e. association of total retinal volume with 1. E/e’ septal and 2. age; or total retinal volume with 1. E/e’ septal and 2. sex). This is now explicitly stated ” We did not include all variables into one model for reasons of statistical power, to ensure a maximum amount of 10% of the sample size as covariate.”  (line 141-143)

Moreover, we agree that we should mention the implications of the unequally distributed sex-distribution in both study groups: “The higher prevalence of females within the HFpEF group compared to control individuals may also have attributed to differences in retinal marker results, although sex-dependent differences are found less prominent7,28,29 than differences between HFpEF and control individuals we observed in the present study. Moreover, sex did not impact the associations we observed between retinal and cardiac diastolic dysfunction markers in patients with HFpEF, suggesting that the retinal alterations are relevant in both sexes.” (line 288-294)

  1. With respect to age there is statistically significant difference in the mean age between HFpEF patients and controls, which can bring the question of registered differences simply as a result of age difference between study subject and controls. Also, other issue is missing data on clinical presentation for control patients, so one cannot exclude the possible influence of hypertension (present in 50% of control patients), or the obesity in controls. Same goes for echocardiography parameters. Having this data would make OCT data more reliable.

Although both groups have significant age differences (median difference of 6 years), the retinal differences found between HFpEF patients and controls is far greater. We now state this more explicitly: ”Age differences between patients with HFpEF and control individuals may have partially attributed to the group differences (median age difference of six years), but the retinal differences between groups exceed expected changes based on reference data for subjects with ages ranging from 30 to 80 years26. This is supported by an observational study reporting, for example, a loss of 0.008 mm3 total retinal volume per year28,which is a hundredfold smaller than our observed group difference of 0.8 mm3.” (line 282-288)

We agree that linear regression analyses in both groups for clinical markers and retinal markers, or adjusted comparative analyses between HFpEF patients and controls, could strengthen the robustness of the findings. However, due to ethical/regulatory restrictions with regard to data from healthy controls, this was not possible for the present study. We clarified this in the manuscript: “Due to ethical and regulatory restrictions, clinical data from control individuals was analyzed anonymously and separately from OCT-A acquisitions, precluding linear regression analyses between clinical data and retinal markers in this group.” (line 148-151)

  1. However, despite methodology shortcomings this study has clinical relevance. Imaging the eye allows a direct view on the retinal arterial and capillary bed and may therefore represent an ideal window to systemic cardiovascular diseases and its microvascular regression and dysfunction. Studies addressing the retinal microvascular components in HFpEF, however, are lacking.

We thank the review for the comments and agree that, although limitations exist, our findings are promising to better understand the cardiac-retinal interaction and may suggest a good non-invasive technique to monitor systemic microvascular dysfunction in cardiovascular disease.

Reviewer 2 Report

Comments and Suggestions for Authors

I have reviewed the manuscript entitled 'Retinal vascular changes in heart failure with preserved ejection fraction by Optical Coherence Tomography Angiography'.

The manuscript is well-presented and points an important issue

AF is an important factor in these patients in order to prevent the progression of HFpEF. Please mention the importance of AF ablation in patients with heart failure. Please consider citing 'Catheter Ablation Approaches for the Treatment of Arrhythmia Recurrence in Patients with a Durable Pulmonary Vein Isolation' and 'Comparison of catheter ablation and medical therapy for atrial fibrillation in heart failure patients: A meta-analysis of randomized controlled trials'.

Please also mention the importance of systemic effects of this disease since the mortality rates of HFpEF patients can sometimes underestimated. The systemic effects of these disease should be emphasized.

Comments on the Quality of English Language

I have reviewed the manuscript entitled 'Retinal vascular changes in heart failure with preserved ejection fraction by Optical Coherence Tomography Angiography'.

The manuscript is well-presented and points an important issue

AF is an important factor in these patients in order to prevent the progression of HFpEF. Please mention the importance of AF ablation in patients with heart failure. Please consider citing 'Catheter Ablation Approaches for the Treatment of Arrhythmia Recurrence in Patients with a Durable Pulmonary Vein Isolation' and 'Comparison of catheter ablation and medical therapy for atrial fibrillation in heart failure patients: A meta-analysis of randomized controlled trials'.

Please also mention the importance of systemic effects of this disease since the mortality rates of HFpEF patients can sometimes underestimated. The systemic effects of these disease should be emphasized.

Author Response

  1. I have reviewed the manuscript entitled 'Retinal vascular changes in heart failure with preserved ejection fraction by Optical Coherence Tomography Angiography'. The manuscript is well-presented and points an important issue

We thank the reviewer for the efforts and are happy to receive a positive reply.

  1. AF is an important factor in these patients in order to prevent the progression of HFpEF. Please mention the importance of AF ablation in patients with heart failure. Please consider citing 'Catheter Ablation Approaches for the Treatment of Arrhythmia Recurrence in Patients with a Durable Pulmonary Vein Isolation' and 'Comparison of catheter ablation and medical therapy for atrial fibrillation in heart failure patients: A meta-analysis of randomized controlled trials'.

We agree with the reviewer that atrial fibrillation (AF) is an important clinical entity without HFpEF, which still warrants further studies to improve clinical management. It has been reported often that AF is a sign of progressive disease in HFpEF, and sinus restoration may improve prognosis in some patients. However, we believe detailed AF discussion falls beyond the scope of the present manuscript. We already included adjusted linear regression analyses for AF status and performed interaction analyses, showing no different results between HFpEF patients with and without AF. (lines 176-178)

Because of the reviewer’s comments, we re-performed comparative analyses of all retinal markers between HFpEF patients with and without AF, finding no retinal OCT-A marker that was different between both groups. Because the latter sub-analyses generate even smaller sample sizes and do not change the conclusions, we did not include them in the manuscript.

  1. Please also mention the importance of systemic effects of this disease since the mortality rates of HFpEF patients can sometimes underestimated. The systemic effects of these disease should be emphasized.

We completely agree with the notion from the reviewer that the systemic effects of HFpEF deserve explicit statements, as well as how this relates to prognosis. We revised the discussion accordingly, which is summarized here: “The present findings are consistent with the hypothesis of HFpEF as a systemic syndrome accompanied by microvascular dysfunction. Microvascular endothelial dysfunction in the heart …… retinal alterations may relate to disease severity in HFpEF and may imply systemic consequences of the syndrome. .... More broadly, the prognostic relevance of retinal disease has been shown in patients with HFpEF and diabetes, in which self-reported retinopathy was associated with future heart failure hospitalizations and higher mortality23”.  (lines 240-256)

Round 2

Reviewer 2 Report

Comments and Suggestions for Authors

Thank you for the required revisions.